# Healthcare Professionals’ Interactions with Families of Hospitalized Patients Through Information Technologies: Toward the Integration of Artificial Intelligence

**DOI:** 10.3390/nursrep15120446

**Published:** 2025-12-12

**Authors:** Jose-Fidencio Lopez-Luna, Ruben Machucho, Frida Caballero-Rico, Ramón Ventura Roque-Hernández, Jorge-Arturo Hernandez-Almazan, Hiram Herrera Rivas

**Affiliations:** 1Information Technologies Department, Polytechnic University of Victoria, Ciudad Victoria 87138, Tamaulipas, Mexico; hherrerar@upv.edu.mx; 2Automotive Systems Department, Polytechnic University of Victoria, Ciudad Victoria 87138, Tamaulipas, Mexico; rmachuchoc@upv.edu.mx; 3University Center, Faculty of Commerce and Administration-Victoria, Autonomous University of Tamaulipas, Ciudad Victoria 87149, Tamaulipas, Mexico; fcaballer@uat.edu.mx; 4Faculty of Commerce, Administration and Social Sciences-Nuevo Laredo, Autonomous University of Tamaulipas, Nuevo Laredo 88000, Tamaulipas, Mexico; rvhernandez@uat.edu.mx

**Keywords:** healthcare professionals, interactions, information technologies, human–AI interaction, artificial intelligence, person-centered care

## Abstract

**Background/Objectives:** The integration of Information Technologies has transformed interactions between healthcare professionals and the families of hospitalized patients, enabling more comprehensive, transparent, and patient-centered care. Artificial Intelligence is emerging as a transformative tool to further enhance these interactions; however, its implementation faces challenges associated with access to and availability of basic technological infrastructure. **Methods:** This cross-sectional pilot study, conducted at the Tamaulipas Children’s Hospital, Mexico, included 51 healthcare professionals from diverse specialties. It examined the use of digital technologies and perceptions of information systems aimed at optimizing communication with families. **Results:** Findings indicated that 58.8% reported consistent use of digital devices, whereas only 41.2% had regular internet access. Between 60.0% and 67.0% consistently provided information regarding patients’ health status, treatments, and medical procedures. With respect to a digital system, 37.3% considered its implementation necessary and 39.2% perceived potential benefits, although functions such as multimedia sharing and automated notifications were regarded with caution. The questionnaire demonstrated high reliability (α = 0.835) and acceptable construct validity (KMO = 0.705; Bartlett’s test *p* < 0.001). **Conclusions:** Preliminary results suggest that the integration of AI-based digital systems in hospital settings remains conditional. They also highlight the need to ensure equitable access to technological infrastructure as a prerequisite for achieving sustainable adoption.

## 1. Introduction

Communication between healthcare professionals and the families of hospitalized patients is a fundamental pillar of patient-centered care and the patient’s environment. This informational link helps to improve patient and family satisfaction, facilitates shared decision-making, reduces relatives’ anxiety, and strengthens confidence in the treatment [1,2]. In recent decades, Information Technologies (IT) have significantly expanded the tools available for this communication, allowing interaction through instant messaging, video calls, electronic portals, and email, among others [2,3]. Furthermore, the increasing integration of Artificial Intelligence (AI) suggests potential benefits, such as partial automation of informational content, personalized alerts, and support in message generation, which could alleviate the workload of healthcare professionals [4,5]. However, the effective use of IT and its combination with AI in this field still face significant challenges. Barriers have been reported related to concerns about data privacy and security, unequal technological accessibility, the availability of internet connections in clinical care areas, the digital skills of staff, and the cultural and ethical acceptance of using automated systems [6,7,8]. AI in health communication tools (chatbots) for family updates runs the risk of spreading misinformation and biases [9]. In turn, case studies reveal misinformation and bias in AI tools that affect family queries regarding incorrect pediatric advice [10]. AI-generated texts for patient summaries risk hallucination and bias propagation [11]. Likewise, there are divergent hypotheses about the extent to which family members prefer face-to-face interaction versus technology-mediated communication, or how much automation should be allowed without compromising accuracy, empathy, and professional judgment [4,12]. Recent studies have shown that healthcare professionals, although they frequently use digital tools for communication support, do not always utilize them consistently and have not fully integrated systems that provide automated or real-time information [13,14]. A study on nurses’ use of IT during the pandemic examined the use of phone calls, video calls, and messages to maintain communication with family members, while also highlighting a significant emotional and operational burden when these resources were not formalized or adequately supported institutionally [15]. Similarly, a study conducted in Latin America reported that physicians frequently used IT tools to communicate with both colleagues and patients; however, they faced significant barriers related to limited time, privacy concerns, and insufficient infrastructure [16].

The present pilot study conducted at the Tamaulipas Children’s Hospital aims to contribute to this emerging field. It investigates how healthcare professionals currently use IT to communicate with family members (health status, treatments, procedures), how frequently and consistently that communication is, and what the perception, acceptance, and expectations are regarding the implementation of an internet-based information system with capabilities (multimedia, automatic notifications, material requests, to mention).

This study provides relevant evidence in a Latin American context that has been little explored in the literature. It highlights clear discrepancies between current practice, expectations, and openness to more automated or AI-assisted systems. It addresses the need for flexible tools that adapt to clinical criteria and preserve fundamental elements of care, such as empathy and personalization. These tools should gradually incorporate automation without eroding the trust or the quality of the interaction between healthcare professionals and the families of hospitalized patients.

This work aimed to characterize the communication practices of healthcare professionals at the Tamaulipas Children’s Hospital with the families of hospitalized patients through information technologies; to assess the frequency and consistency of these interactions; and to identify perceptions regarding the implementation of an online information system with advanced features, as a preliminary step toward its integration with Artificial Intelligence.

## 2. Materials and Methods

The study was a descriptive, cross-sectional pilot in two phases [17]; through the application of surveys to 51 health professionals at the Tamaulipas Children’s Hospital, the state’s main pediatric center, serves nearly 168,000 people and offers care in 34 medical specialties or subspecialties and eight clinics, supported by a multidisciplinary team that promotes comprehensive care and research in child health. It should be noted that the study was conducted during the Severe Acute Respiratory Syndrome Coronavirus 2 (SARS-CoV-2) pandemic, which posed difficulties in recruiting participants for the study [18], and the main barriers were health restrictions and protocols, staff burnout, and turnover [19]. The sampling was subject to the availability of the respondents’ workload. This made it impossible to apply a probabilistic procedure. Therefore, convenience sampling was chosen, a strategy frequently used in exploratory, preliminary and pilot studies [20]. The instrument used in this study comprises three latent dimensions—Use of Information Technologies, Information Needs, and Interactions—which were treated as the primary analytical variables [21]. The ratios between the sample and the variable were 5:1 and 15:1 [22]. Reliability validation was conducted using Cronbach’s Alpha coefficient [23], and construct validation was performed through factor analysis [24,25,26].

The data were collected using Google Sheets online, stored and coded in Microsoft Excel 2016, and statistically processed with the Statistical Package for the Social Sciences (SPSS) version 25.0.

Ethical considerations: The research was classified as “risk-free research,” as it consisted of administering a questionnaire without considering sensitive aspects of the behavior of the participating individuals, nor any intentional intervention or change in their physiological, psychological, or social variables. Institutional Review Board Statement: The study was conducted in accordance with the Declaration of Helsinki [27], and approved by the Research Ethics Committee of Tamaulipas Children’s Hospital Committee (protocol code: HIT-INV-2022-3 and approval date: 27 June 2022). Digital informed consent was obtained from participants who were willing to participate [28,29]. The survey was anonymous and confidential.

Inclusion, exclusion, and elimination criteria: To meet the inclusion criterion, health professionals had to be affiliated with Tamaulipas Children’s Hospital. The exclusion criterion was that they could not be located due to work reasons or had workloads that prevented them from responding to the survey request. Finally, the elimination criteria for participants were: incomplete surveys or individuals who decided to withdraw from the study.

Instrument: For data collection, a preliminarily validated survey entitled “Interactions through Information Technologies for Health Professionals” was used at the Tamaulipas Children’s Hospital includes a text referring to informed consent and consists of an initial question asking the respondent whether they have read, understood, and agree to participate in the study; if their answer is affirmative, we proceed with four sections of questions: Demographic Data (adapted to the interests of the Tamaulipas Children’s Hospital) with five questions, Use of Information Technologies with four questions, Information Needs with four questions, and Interactions (willingness) with six questions [21].

Procedure: The research project protocol was submitted to the Division of Research, Quality, and Planning at Tamaulipas Children’s Hospital, which referred the proposed protocol for evaluation by the Tamaulipas Children’s Hospital Research Ethics Committee, Registration Number: CONBIOETI-CA-28-CEI-001-20190218, and to the Committee on Research, Teaching, and Library. Once the committees accepted the research protocol, the Hospital Management issued a letter to Division and Specialty heads, requesting their support in inviting the staff assigned to their areas to participate in the aforementioned project voluntarily. Health professionals who accepted the invitation accessed the survey by entering the web address manually or via a QR code, completing it individually and without any supervision or assistance.

## 3. Results

Out of a total of 51 healthcare professionals participating in the pilot study conducted at Tamaulipas Children’s Hospital, 32 (62.7%) were women, representing the higher percentage, and 19 (37.3%) were men. Among the participants, the age range of 30 to 39 years was the most common, representing a significant portion of the total at 37.3% (19 healthcare professionals). Likewise, the data obtained indicate that 49 (96.1%) of the participants reside in urban areas. It is also noteworthy that the profession of specialist physician stands out, accounting for 19 (37.3%) of the counts, followed by the profession of psychologist with 12 (23.5%), while other professions accounted for 2 (3.9%) of the counts. Finally, the hospitalization department and the “other” option account for 12 (23.5%) of the counts, respectively. Detailed information on the demographic data is presented in Table 1.

Regarding the use of information technologies reported by healthcare professionals, only 41.2% “always” use the Internet in their daily professional practice. It is notable that 58.8% “always” use a computer, tablet, or smartphone for their daily professional practice. It is striking that 23.5% indicate “almost always”, and the availability of an Internet connection at the workplace, where these devices are used, is “only sometimes”. Finally, 37.3% report that they “always” use various communication platforms, such as SMS messages, email, WhatsApp, Facebook, Twitter, among others, for their daily professional practice, see Figure 1.

Regarding the information provided by healthcare professionals to meet the informational needs of the families of hospitalized patients, it is noted that 60.8% of respondents “always” inform the families about the health status of their hospitalized patient, 64.7% “always” inform the families about how their hospitalized patient is being medically treated, while 60.8% “always” accurately communicate to the family what is being done medically to the hospitalized patient, and finally, 66.7% “always” inform the families about the reasons for the procedures performed on their hospitalized patient, see Figure 2.

In terms of how healthcare professionals perceive the use of an Internet-based information system across various devices such as computers, tablets, and smartphones, including the ability to communicate via SMS and social media, 37.3% of healthcare professionals “always” stated the need to implement a new information system; 39.2% indicated that it would “always” be beneficial to use a new information system. It is noteworthy that 45.1% “only sometimes” think it is necessary to provide multimedia content (photos and/or videos) through the use of an information system, 33.3% answered that they “only sometimes” find it necessary to make requests for materials (personal hygiene items, clothing, medications, etc.) or documentation. In comparison, 41.2% stated that they would “only sometimes” use a system that incorporates both communication capabilities and informational functions. Finally, 39.2% are “only sometimes” interested in reporting the follow-up of hospitalized patients through the use of an information system that automatically notifies the patient’s family via electronic messages to their preferred social networks, as shown in Figure 3.

Finally, it is worth mentioning that a Cronbach’s Alpha of 0.835, indicating excellent reliability, was obtained for the questionnaire. Regarding the reliability of the dimensions, Cronbach’s Alphas of 0.798, 0.960, and 0.859 were obtained for the use of information technologies, information needs, and interactions, respectively. Concerning validity, a moderate value of 0.705 was obtained for the Kaiser-Meyer-Olkin (KMO) measure, and Bartlett’s test of sphericity was significant (Chi-square = 561.872, df = 91, Sig. < 0.001).

## 4. Discussion

In this pilot study, the communication practices of healthcare professionals at the Tamaulipas Children’s Hospital with the families of hospitalized patients were characterized through the use of information technologies, evaluating both the frequency and consistency of these interactions. Likewise, perceptions regarding the possible implementation of an online information system with advanced functionalities were explored, conceived as a preliminary stage toward its future integration with Artificial Intelligence tools. The findings obtained allow for outlining an initial overview of how these practices are carried out in the hospital context, identifying relevant patterns and areas of opportunity that could guide subsequent, more extensive research with a focus on the technological optimization of clinical communication, including the implementation of Artificial Intelligence.

In this preliminary study, a limited use—below 50%—of information technologies was identified, along with a low willingness among healthcare professionals to adopt new communication systems. This scenario suggests that a lack of familiarity with IT and the perception that its incorporation may complicate workflows or increase workload negatively influence adoption, while organizational factors such as insufficient training, limited technical support, and a lack of institutional incentives also appear to reinforce this resistance [30]. Nevertheless, the quality of the information provided exceeded the average expectations. This fact demonstrates an untapped potential: current systems offer clinical value, but their impact depends on implementation strategies that improve usability, strengthen digital skills, and highlight practical benefits for healthcare professionals [31].

Overall, the study revealed that, in their daily professional practice, 41.2% of healthcare professionals use the Internet, 58.8% use devices such as computers, tablets, or smartphones, 23.5% have Internet access at their workplace to use these devices, and 37.3% report using electronic messages. These findings align with previous research showing limited computer skills and usage among healthcare professionals, particularly in primary care. For example, a cross-sectional study in Ethiopia involving 554 health professionals across healthcare facilities found similar results regarding knowledge, computer use, and related factors [32]. However, in the present study, the participants belong to a tertiary-level hospital institution, which represents a different context. The low percentage of information technology use by professionals could be attributed to the fact that their main priority remains direct patient care rather than the use of technological tools.

This study also highlights the level of information that healthcare professionals provide to the families of hospitalized patients. 60.8% of respondents communicate the patient’s health status, 64.7% inform about the medical treatment being applied, 60.8% explain precisely what is being done medically, and 66.7% indicate the reasons for the procedures performed. It is worth noting that healthcare professionals may need to offer extra information to families without Internet access and help verify or clarify online health information for those who use it [33]. The high level of information communicated by medical staff can be considered a key component of patient-centered care.

The health professionals’ perceptions regarding the potential implementation of a new information system. 37.3% of the respondents considered such implementation necessary; 39.2% stated that it would be beneficial; 45.1% indicated the need to include multimedia content (photos and/or videos); 33.3% considered it important to be able to request materials (such as hygiene items, clothing, or medications) or documentation; 41.2% indicated they would use the system for communication functions; and 39.2% expressed interest in the system allowing them to report on the follow-up of hospitalized patients, including automatic notifications to family members via electronic messages. However, evidence from other contexts is encouraging. For example, a 2023 mixed-methods study conducted in three hospitals in Ghana found that most professionals viewed Electronic Medical Records positively, with over 80.0% believing that these systems benefited patients and approximately 75.0% reporting improvements in workflows and clinical outcomes [34].

In international contexts, the adoption of digital tools may be facilitated by more robust health policies and more advanced technological infrastructures. In contrast, in developing countries like Mexico, limited basic healthcare can hinder the adoption of such systems, in addition to personnel obstacles, including limited digital skills, high turnover, resistance to change, reluctance to adopt quality-focused practices, and low participation [35], and institutional obstacles such as technological integration issues, lack of resources, and leadership support [36,37], as well as organizational structures that hinder the implementation of improvements.

Below, we discuss the findings, highlighting certain aspects of the interactions between healthcare professionals and the families of hospitalized patients from the perspective of healthcare staff, specifically regarding the use of information technologies, information needs, and interactions.

37.3% of healthcare professionals reported using electronic messaging “always” in their daily professional practice, a figure similar to the 32.0% of Argentine pediatricians who agreed with the benefits of this tool in communication with their patients [38]. Likewise, a study in Israel found that healthcare professionals viewed SMS as a convenient, easy-to-use tool for updating parents on their premature babies’ health [39]. Which suggests a global trend in healthcare where technological advances are reshaping interactions among professionals, patients, families, and care teams.

On the other hand, the results showed that over 60% of healthcare professionals met families’ information needs regarding patients’ conditions, treatments, and procedures, likely reflecting the increasing demand for transparency in healthcare. Similarly, studies with relatives of hospitalized patients highlight that receiving clear information is essential for coping [40] and is among their top informational priorities [41].

Regarding the perceptions of the need for healthcare professionals to interact with patients’ families through a two-way information system, accessible via the Internet on a variety of technological devices with automatic sending of electronic messages, it is reported as a need “always” by 37.3% of respondents and would be beneficial to them “almost all the time”. In this regard, a systematic review identified 850 studies on how Electronic Health Records support communication among patients, families, and healthcare professionals, but found few that directly addressed this interaction, concluding that such systems are still rarely implemented in clinical settings [42]. The literature also indicates that information systems accessible from devices are becoming an integral part of modern medical systems, enhancing accessibility, efficiency, and potentially improving the quality of healthcare [43]. The interest in multimedia content stems from its ability to enhance understanding of a hospitalized patient’s condition, as images or videos provide clearer, more tangible information than text or verbal explanations [44]. However, the percentage of healthcare professionals increases to 45.1% who consider it necessary to provide multimedia content (photos and/or videos) through the use of an information system, while the frequency decreases to “only occasionally”. This variation in perceptions may be due to differences in the type of patients treated or the severity of the conditions.

In the healthcare field, AI stands out for its usefulness in data analysis and decision-making support, enhancing the quality of care, diagnostic efficiency, and administrative management through programming systems and alerts [45,46,47]. Doctors and nurses find AI tools very useful, such as diagnostic assistants, predictive analytics, automated documentation systems, and clinical alerts, for improving efficiency and reducing workload [47,48]. Furthermore, AI can transform communication in the hospital setting through tools such as chatbots and virtual health assistants, which facilitate information exchange and remote monitoring [49,50]. These platforms allow for automatic updates and alerts regarding the patient’s status or care plans, reducing misinformation, promoting transparency, and encouraging family involvement in the care process [51].

Consequently, it is recommended to move towards the consolidation of innovative models of clinical communication mediated by information technologies and enhanced by artificial intelligence [52], considering that artificial intelligence systems are still unable to reason in the same way as human healthcare professionals, who can rely on common sense or on intuition and experience when interacting with the families of hospitalized patients [53].

The comparative analysis of both studies indicates that both the preliminary validation [21] and this study demonstrate excellent internal consistency for the questionnaire. Although the pilot study shows slightly lower values of overall reliability and KMO, it also reflects an improvement in reliability in two of the three dimensions (information needs and interactions). The consistent significance in Bartlett’s test in both studies supports the construct validity of the instrument. The slight decrease in the reliability of the use of information technologies factor when moving from simulated healthcare professionals to real healthcare professionals reflects the complexity of the real environment, where perceptions and experiences are more diverse and less standardized [54,55].

Regarding the study’s limitations, it is noted that the sample size of 51 healthcare professionals does not constitute a representative sample. However, due to practical considerations and the number of available participants, the sample size is considered sufficient in this type of study to validate the results adequately. Among the limitations of this study is the use of non-probabilistic sampling, which means that the sample may not fully reflect the characteristics of the total population of healthcare professionals at the Tamaulipas Children’s Hospital (HIT). This condition could introduce a certain degree of bias and limit the generalization of the results to other contexts or institutions. Finally, it is relevant to validate the results found at a multicenter level, in future studies, and to delve into what prevents healthcare professionals from connecting to the Internet. Additionally, it is important to investigate what patients’ families think about the use of technology to receive updates on their hospitalized loved ones, through the application of the corresponding instrument.

## 5. Conclusions

This pilot study contributes empirical evidence to the knowledge about communication practices of healthcare professionals with the families of hospitalized patients through information technologies, highlighting both the frequency and consistency of such interactions as well as perceptions regarding the possible implementation of advanced systems integrated with artificial intelligence.

The findings suggest that, although technology-mediated communication represents a strategic tool to strengthen the relationship between healthcare professionals and hospitalized patient families, its effectiveness critically depends on institutional support, the formalization of protocols, and the cultural adaptation of the strategies employed, to pro-mote family-centered care.

This study can serve as a basis for the development of training programs aimed at strengthening the communicative and technological competencies of healthcare professionals, from an integrative perspective that is sensitive to organizational. Likewise, the findings offer implications for the design of institutional policies that promote the incorporation of digital tools and intelligent systems as key components to improve the quality and humanization of care.

On a more general level, this perspective points to the possibility of exploring care environments that gradually and systematically incorporate communication technologies and artificial intelligence-assisted systems, to support specific aspects of human-digital interaction and contribute, in a complementary way, to the care processes that involve patients, their families and health professionals.

## Figures and Tables

**Figure 1 nursrep-15-00446-f001:**
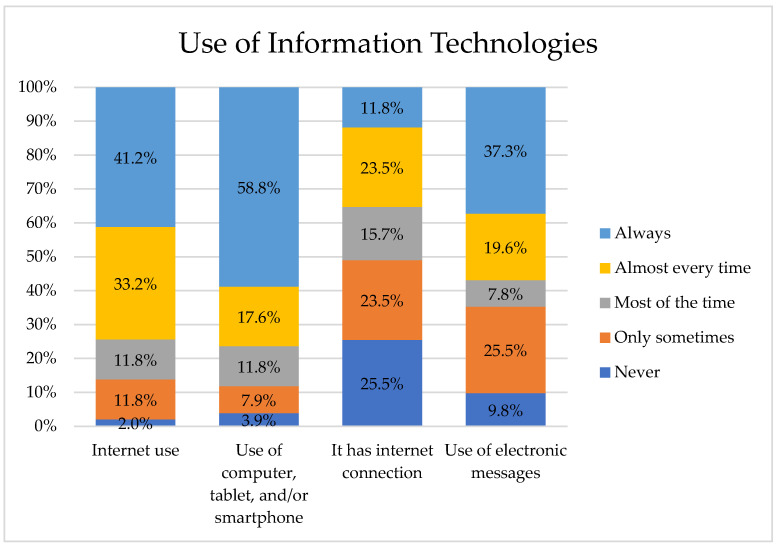
Health professionals who incorporate IT in their professional work.

**Figure 2 nursrep-15-00446-f002:**
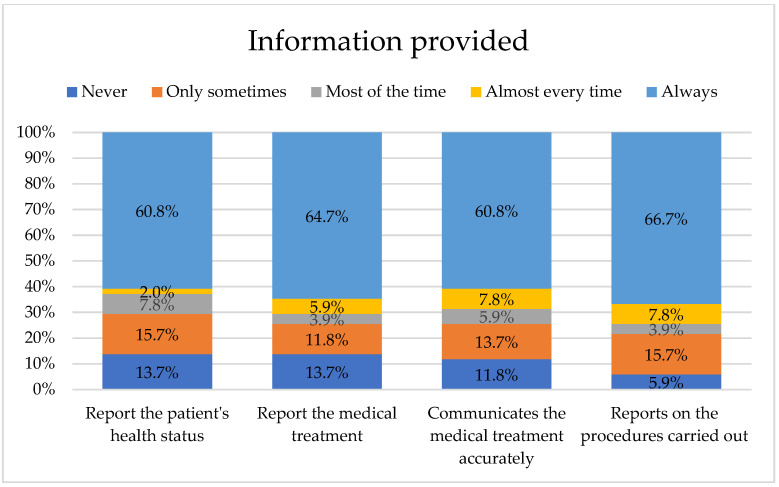
Information provided by health professionals to the relatives of hospitalized patients.

**Figure 3 nursrep-15-00446-f003:**
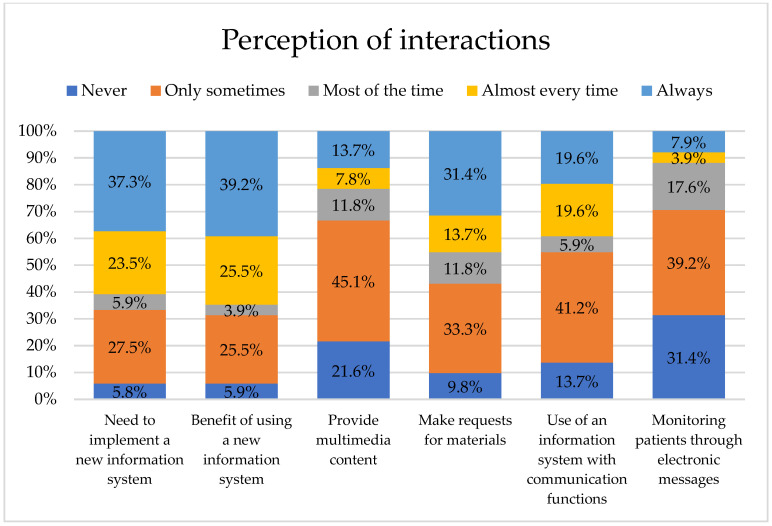
Professionals’ perceptions of an online information system with advanced features.

**Table 1 nursrep-15-00446-t001:** Demographic data of the participants.

Demographic Data	Value	Count	Percentage
Sex	Man	19	37.3%
	Woman	32	62.7%
	Total	51	100.0%
Age	From 18 to 29 years old	6	11.8%
	From 30 to 39 years old	19	37.3%
	From 40 to 49 years old	15	29.4%
	From 50 to 59 years old	8	15.7%
	From 60 to 69 years old	2	3.9%
	Over 69 years	1	2.0%
	Total	51	100.0%
Place of residence	Rural	2	3.9%
	Urban	49	96.1%
	Other	0	0.0%
	Total	51	100.0%
Profession	Social worker	0	0.0%
	Nurse	5	9.8%
	Specialist doctor	19	37.3%
	Resident doctor	10	19.6%
	Psychologist	12	23.5%
	Nutritionist	3	5.9%
	Dentist	0	0.0%
	Other	2	3.9%
	Total	51	100.0%
Area or department	Neonatal intensive care unit	6	11.8%
	Pediatric intensive care unit	4	7.8%
	Hospitalization	12	23.5%
	Emergency	6	11.8%
	Surgery	0	0.0%
	Oncology and hematology	10	19.6%
	Administrative	1	2.0%
	Other	12	23.5%
	Total	51	100.0%

## Data Availability

The data presented in this study are available from the author upon request due to privacy considerations.

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
