# Peer review of "Healthcare Professionals’ Interactions with Families of Hospitalized Patients Through Information Technologies: Toward the Integration of Artificial Intelligence"

_nursrep, 2025, doi:10.3390/nursrep15120446_

Round 1
Reviewer 1 Report
Comments and Suggestions for Authors
Summary:
Researchers looked at how doctors and nurses at a Mexican children’s hospital use technology - like computers and apps - as well as their thoughts on new AI tools to keep families informed about patients. They surveyed 51 staff members to understand what they do now, alongside their feelings about adopting more sophisticated systems.
Strengths:
1. This research looks at how technology - specifically computers and artificial intelligence - is changing conversations within healthcare, especially when families are involved in patient care. It feels important right now.
2. Researchers detailed their methods, securing necessary ethics clearances alongside proper consent from participants.
3. Scores on the survey were consistent - a reliability check showed 0.835. Moreover, results indicated that it measured what it intended to, with a KMO value of 0.705.
4. Findings show up plainly, bolstered by charts likewise data displays.
5. This work thoughtfully places its results alongside what others have already discovered.
Weaknesses and Recommendations:
1. With only 51 healthcare workers from one hospital participating, the results might not apply elsewhere. The researchers admit this makes broad conclusions tricky. To strengthen the work, they could involve more people at various hospitals - or repeat the study on a bigger scale.
2. It’s a bit worrying that just over 40% of doctors and nurses consistently go online; this could make using technology to improve things tricky. We really need to dig deeper - what exactly stops people from getting connected? Does it differ depending on where they work within the hospital, or when they’re on duty?
3. Access to tech isn’t equal, this research shows. We should talk more about fairness when building new ways to communicate digitally - making certain families lacking consistent internet aren’t left behind.
4. The research points to AI being a big deal, yet doesn’t detail exactly which AI capabilities matter most. So, what precise AI tools do doctors and nurses find helpful? Moreover, how do they picture AI improving talks with patients’ relatives?
5. Right now, we only hear from doctors and nurses. To build truly useful tools, we also need to understand what patients’ families think about using tech for updates – whether it fits how they want things done.
6. Though over a third recognize the need for this - nearly 40% foresee advantages - the research barely touches on what might actually prevent its use, aside from technical issues. What obstacles within companies, their habits, or staff skills could stand in the way?
Minor Comments:
To really paint a picture, let’s flesh out where this happens - think about how big the hospital is, how many people they treat, also what kinds of medical experts work there. This gives folks a better sense of the scene.
Let’s talk about how many people replied, also whether those who didn’t might skew the results.
Where helpful, show uncertainty around numbers using error bars or ranges.
Conclusion:
The initial research offers useful first thoughts on how doctors view tech - both regular IT alongside artificial intelligence - when talking with families. It shows promise, yet also real hurdles, especially getting everyone reliable access. With some revisions – notably, more detail about what gets in the way of using this technology and ensuring fair access for all – publishing seems like a good next step.
Author Response
Healthcare Professionals’ Interactions with Families of Hospitalized Patients Through Information Technologies: Toward the Integration of Artificial Intelligence. Manuscript ID: nursrep-3945818
|
Response to Reviewer 1 Comments
|
||
|
Summary |
|
|
|
Thank you very much for taking the time to review this manuscript. Please find the detailed responses below and the corresponding revisions/corrections highlighted.
|
||
|
Point-by-point response to Comments and Suggestions for Authors |
||
|
Comments 1: With only 51 healthcare workers from one hospital participating, the results might not apply elsewhere. The researchers admit this makes broad conclusions tricky. To strengthen the work, they could involve more people at various hospitals - or repeat the study on a bigger scale. |
||
|
Response 1: Thank you for pointing this out. We agree with this comment. Therefore, we have addressed your comment, it is highlighted that the study was conducted during the pandemic (SARS-CoV-2), in the discussion section, page eight, paragraph six and line 305. Text added to the manuscript: "It should be noted that the study was conducted during the Severe Acute Respiratory Syndrome Coronavirus 2 (SARS-CoV-2) pandemic, which posed difficulties in recruiting participants for the study [49], and the main barriers were health restrictions and protocols, staff burnout, and turnover [50]." Future work with larger samples is also indicated, page nine, last paragraph of the limitations in the discussion section and line 325. Text added to the manuscript “Finally, it is relevant to validate the results found at a multicenter level, in future studies, and to delve into what prevents healthcare professionals from connecting to the Internet. Additionally, it is important to investigate what patients' families think about the use of technology to receive updates on their hospitalized loved ones, through the application of the corresponding instrument.” Comments 2: It’s a bit worrying that just over 40% of doctors and nurses consistently go online; this could make using technology to improve things tricky. We really need to dig deeper - what exactly stops people from getting connected? Does it differ depending on where they work within the hospital, or when they’re on duty? Response 2: Thank you for pointing this out. We agree with this comment. Therefore, we have addressed your comment, specifying that it is part of the future work to be done, is also indicated, page nine, last paragraph of the limitations in the discussion section and line 325. Text added to the manuscript “Finally, it is relevant to validate the results found at a multicenter level, in future studies, and to delve into what prevents healthcare professionals from connecting to the Internet. …” Comments 3: Access to tech isn’t equal, this research shows. We should talk more about fairness when building new ways to communicate digitally - making certain families lacking consistent internet aren’t left behind. Response 3: Thank you for pointing this out. We agree with this comment. Therefore, we have addressed your comment, specifying that it is part of the future work to be done, is also indicated, page nine, last paragraph of the limitations in the discussion section and line 327. Text added to the manuscript “… Additionally, it is important to investigate what patients' families think about the use of technology to receive updates on their hospitalized loved ones, through the application of the corresponding instrument.” Comments 4: The research points to AI being a big deal, yet doesn’t detail exactly which AI capabilities matter most. So, what precise AI tools do doctors and nurses find helpful? Moreover, how do they picture AI improving talks with patients’ relatives? Response 4: Thank you for pointing this out. We agree with this comment. Therefore, we have addressed each of the questions in your comment in the page eight, paragraph four in the discussion section and line 289. Text added to the manuscript “In the healthcare field, AI stands out for its usefulness in data analysis and decision-making support, enhancing the quality of care, diagnostic efficiency, and administrative management through programming systems and alerts [42-44]. Doctors and nurses find AI tools very useful, such as diagnostic assistants, predictive analytics, automated documentation systems, and clinical alerts, for improving efficiency and reducing workload [44,45]. Furthermore, AI can transform communication in the hospital setting through tools such as chatbots and virtual health assistants, which facilitate information exchange and remote monitoring [46,47]. These platforms allow for automatic updates and alerts regarding the patient's status or care plans, reducing misinformation, promoting transparency, and encouraging family involvement in the care process [48].” Comments 5: Right now, we only hear from doctors and nurses. To build truly useful tools, we also need to understand what patients’ families think about using tech for updates – whether it fits how they want things done. Response 5: Thank you for pointing this out. We agree with this comment. Therefore, we have addressed your comment, currently, the scope of this study is focused on healthcare professionals. However, as part of future work, the corresponding study will be conducted focusing on the family members of hospitalized patients. It is also indicated in page nine, last paragraph of the limitations in the discussion section and line 327. Text added to the manuscript “… Additionally, it is important to investigate what patients' families think about the use of technology to receive updates on their hospitalized loved ones, through the application of the corresponding instrument.” Comments 6: Though over a third recognize the need for this - nearly 40% foresee advantages - the research barely touches on what might actually prevent its use, aside from technical issues. What obstacles within companies, their habits, or staff skills could stand in the way? Response 6: Thank you for pointing this out. We agree with this comment. Therefore, we have addressed your comment, we add personnel obstacles and institutional obstacles in page seven, paragraph five in the discussion section and line 250. Text added to the manuscript “…in addition to personnel obstacles, including limited digital skills, high turnover, resistance to change, reluctance to adopt quality-focused practices, and low participation [32], and institutional obstacles such as technological integration issues, lack of resources, and leadership support [33,34], as well as organizational structures that hinder the implementation of improvements. Minor Comments: To really paint a picture, let’s flesh out where this happens - think about how big the hospital is, how many people they treat, also what kinds of medical experts work there. This gives folks a better sense of the scene. Let’s talk about how many people replied, also whether those who didn’t might skew the results. Where helpful, show uncertainty around numbers using error bars or ranges. Response to minor comments: Thank you for pointing this out. We agree with this comment. Therefore, we have… addressed your comment, to really paint a picture we added in the materials and methods section, page three, paragraph one and line 93. Text added to the manuscript “the state's main pediatric center, serves nearly 168,000 people and offers care in 34 medical specialties or subspecialties and eight clinics, supported by a multidisciplinary team that promotes comprehensive care and research in child health.” We have also added respect to outcome bias in the limitations in the discussion section, page eight, last paragraph and line 321. Text added to the manuscript “Among the limitations of this study is the use of non-probabilistic sampling, which means that the sample may not fully reflect the characteristics of the total population of healthcare professionals at the Tamaulipas Children's Hospital (HIT). This condition could introduce a certain degree of bias and limit the generalization of the results to other contexts or institutions.” |
||

Reviewer 2 Report
Comments and Suggestions for Authors
Dear Authors
This study is an original pilot research that examines the communication processes between healthcare professionals and the families of hospitalized patients in the context of information technologies, while also discussing how these interactions may evolve toward the integration of artificial intelligence in the future. The topic is timely and relevant, contributing to current debates on the digital transformation of patient- and family-centered care. It carries importance from both theoretical and practical perspectives.
Introduction: The introduction clearly defines the scope, rationale, and objectives of the study. The literature cited is up-to-date and relevant, addressing both technological and ethical/cultural dimensions, which gives the work a comprehensive character. Including a brief mention of the potential risks of AI integration (e.g., misinformation, data bias) would provide a more balanced view. The expression “pilot study” is repeated several times; while the emphasis can be retained, the repetition may be reduced for smoother readability.
Methods: The methodology is clearly structured, systematic, and ethically sound. The section reflects the transparency and validation steps required for a pilot study. Although convenience sampling is acceptable for a pilot design, it should be noted that this choice may introduce potential bias — for instance, excluding professionals with heavy workloads may limit representativeness. It is not specified whether the survey was pretested (e.g., through a preliminary pilot or expert validation). Including this information would strengthen methodological reliability. Overall, the methods section contains an appropriate level of detail for a pilot study and serves as a model in terms of ethical rigor. However, a more detailed discussion of sample limitations and the validation process of the survey instrument would enhance methodological transparency.
Results: The results section is well organized, data-driven, and aligned with the study’s objectives.
Discussion: The discussion is coherent with the findings, supported by literature, and contextually appropriate. Despite the limited scope typical of a pilot study, it successfully articulates both theoretical and practical contributions. The literature review is strong; however, the section is text-dense, and adding brief summarizing or transitional paragraphs could improve readability.
Conclusion: The article lacks a separate “Conclusion” section. Although the discussion includes conclusion-like statements, a distinct section should be added to comply with journal formatting standards and to clearly summarize key findings, implications, and future directions.
References: The literature base is adequate but could be strengthened by incorporating more recent works (2023–2025) on digital health communication and artificial intelligence applications.
Language and Terminology: The terms “specialist physician” and “specialist doctor” are used interchangeably; consistent terminology is recommended.
Author Response
Healthcare Professionals’ Interactions with Families of Hospitalized Patients Through Information Technologies: Toward the Integration of Artificial Intelligence. Manuscript ID: nursrep-3945818
|
Response to Reviewer 2 Comments
|
||
|
1. Summary |
|
|
|
Thank you very much for taking the time to review this manuscript. Please find the detailed responses below and the corresponding revisions/corrections highlighted/in track changes in the re-submitted files.
|
||
|
Point-by-point response to Comments and Suggestions for Authors |
||
|
Comments 1: Including a brief mention of the potential risks of AI integration (e.g., misinformation, data bias) would provide a more balanced perspective. The phrase “pilot study” is repeated several times; while the emphasis can be maintained, the repetition could be reduced for a smoother reading experience. |
||
|
Response 1: Thank you for pointing this out. We agree with this comment. Therefore, we have addressed your comment including a brief mention of the potential risks of AI integration. in page two, paragraph one in the introduction section and line 54. Text added to the manuscript “AI in health communication tools (chatbots) for family updates runs the risk of spreading misinformation and biases [10]. In turn, case studies reveal misinformation and bias in AI tools that affect family queries regarding incorrect pediatric advice [11]. AI-generated texts for patient summaries risk hallucination and bias propagation [12].” We have also shortened the expression "pilot study" to achieve a smoother reading throughout the document. Comments 2: Methods: It is not specified whether the survey was pretested (e.g., through a preliminary pilot or expert validation). Including this information would strengthen methodological reliability. A more detailed discussion of the sample limitations and the survey instrument validation process would improve methodological transparency. Response 2: Thank you for pointing this out. We agree with this comment. Therefore, we have addressed your comment, specifying that it is a preliminarily validated survey in page three, paragraph five in the materials and methods section and line 120. Text added to the manuscript” For data collection, a preliminarily validated survey entitled” …. “was used at the” …. In this regard, a more detailed discussion on the limitations of the sample and the validation process of the survey instrument would improve methodological transparency. We have addressed your comment with two paragraphs of text on the matter, The first paragraph in page eight, paragraph six in the discussion section and line 305. Text added to the manuscript “It should be noted that the study was conducted during the Severe Acute Respiratory Syndrome Coronavirus 2 (SARS-CoV-2) pandemic, which posed difficulties in recruiting participants for the study [49], and the main barriers were health restrictions and proto-cols, staff burnout, and turnover [50].” And the second paragraph in pages eight and nine, last and first paragraph respectively in the discussion section and lines 309 to 317. Text added to the manuscript “The comparative analysis of both studies indicates that both the preliminary validation [28] and this study demonstrate excellent internal consistency for the questionnaire. Although the pilot study shows slightly lower values of overall reliability and KMO, it also reflects an improvement in reliability in two of the three dimensions (information needs and interactions). The consistent significance in Bartlett's test in both studies supports the construct validity of the instrument. The slight decrease in the reliability of the use of in-formation technologies factor when moving from simulated healthcare professionals to real healthcare professionals reflects the complexity of the real environment, where perceptions and experiences are more diverse and less standardized [51,52].” Comments 3: The discussion section is text-dense, and adding short summary or transition paragraphs could improve readability. Response 3: Thank you for pointing this out. We agree with this comment. Therefore, we have…. converted long paragraphs into shorter or transitional paragraphs to improve readability in the discussion section. These are listed below: In page seven, second paragraph in the discussion section and line 218. Text added to the manuscript “These findings align with previous research showing limited computer skills and usage among healthcare professionals, particularly in primary care. For example, a cross-sectional study in Ethiopia involving 554 health professionals across healthcare facilities found similar results regarding knowledge, computer use, and related factors [29].” In page seven, paragraph three in the discussion section and line 230. Text added to the manuscript “It is worth noting that healthcare professionals may need to offer extra information to families without Internet access and help verify or clarify online health information for those who use it [30].” In page eight, paragraph one in the discussion section and line 262. Text added to the manuscript “Likewise, a study in Israel found that healthcare professionals viewed SMS as a convenient, easy-to-use tool for updating parents on their premature babies’ health [36]. Which suggests a global trend in healthcare where technological advances are reshaping interactions among professionals, patients, families, and care teams.” In page eight paragraph two in the discussion section and line 266. Text added to the manuscript “On the other hand, the results showed that over 60% of healthcare professionals met families’ information needs regarding patients’ conditions, treatments, and procedures, likely reflecting the increasing demand for transparency in healthcare. Similarly, studies with relatives of hospitalized patients highlight that receiving clear information is essential for coping [37] and is among their top informational priorities [38].” In page eight, paragraph three in the discussion section and line 275. Text added to the manuscript “In this regard, a systematic review identified 850 studies on how Electronic Health Records support communication among patients, families, and healthcare professionals, but found few that directly addressed this interaction, concluding that such systems are still rarely implemented in clinical settings [39].” In page eight, paragraph three in the discussion section and line 281. Text added to the manuscript “The interest in multimedia content stems from its ability to enhance understanding of a hospitalized patient’s condition, as images or videos provide clearer, more tangible in-formation than text or verbal explanations [41].” |
||

Reviewer 3 Report
Comments and Suggestions for Authors
A few questions / comments and suggestions:
The title, abstract, introduction, and conclusion heavily emphasize “Artificial Intelligence”. However, the study itself does not implement, test, or directly measure any aspect of AI. The survey questions pertain to a potential ‘internet-based information system’ with features like multimedia sharing and automated notifications. While these could be ‘components’ of an AI-driven system, they are not inherently AI. The manuscript consistently makes a logical leap from the findings about a basic IT system to broad conclusions about AI integration. Lines 348-351, the conclusion states, “...this vision promotes the development of care ecosystems supported by artificial intelligence... capable of strengthening human-digital interaction...” This is a significant overstatement. The data, which shows low internet access (41.2%) and lukewarm interest in a new system (only ~39% see it as beneficial), points more to fundamental barriers to basic IT adoption than to the readiness for sophisticated AI ecosystems. Suggests the authors should significantly tone down the AI-related claims throughout the manuscript to more accurately reflect the study's scope. The paper is about ‘perceptions of an advanced IT system as a preliminary step toward potential AI integration’. The framing should be adjusted to manage reader expectations and ensure the conclusions are directly supported by the data presented.
Lines 79-82, the sentence, “Importantly, it underscores the need for flexible tools that respect clinical judgment, empathy, and personalization”, sounds like a finding or a conclusion. The introduction should set the stage and state the research question, not pre-emptively declare the study’s findings. Suggests this sentence should be rephrased or moved to the Discussion/Conclusion section, where it would be appropriate to interpret the study’s results.
Lines 97-100, the justification for the sample size is weak. Stating it is “based on the number of participants in pilot studies [19]” is vague. While the mention of a 5:1 to 15:1 ratio of sample-to-variable is a known rule of thumb for factor analysis, the authors do not state how many variables were used in their analysis to demonstrate that this ratio was met. Suggests: while this is a pilot study and a small sample is expected, the justification could be clearer. More importantly, the limitations section already correctly notes the small N, which mitigates this issue to some extent. However, the initial justification in the methods section should be phrased more cautiously.
Line 140-198, results section, the results are reported with two decimal places (e.g., 58.80%, 37.30%). For a small sample size of N=51, this implies a level of precision that does not exist. For instance, 19 out of 51 participants is 37.2549...%. Reporting this as 37.30% is statistically inappropriate. Suggests percentages should be rounded to one decimal place (e.g., 37.3%) or to the nearest whole number (e.g., 37%), which is standard and more honest for this sample size.
Line 162-163, 173-174, 190-191, Figures 1, 2, and 3 are highly problematic and present the data in a misleading way. They selectively display only one response category (e.g., “Always” or “Only sometimes”) for each question, omitting the rest of the response distribution (e.g., “Almost always”, “Almost never”, “Never”). This practice is inappropriate as it hides the full context of the participants’ responses and can lead to incorrect interpretations. For example, in Figure 1, stating that 41.20% “always” use the internet does not tell the reader what the other 58.80% do. Suggests the bar charts must be revised to show the full distribution of responses for each item on the Likert scale. A stacked bar chart or grouped bar chart showing the percentages for all categories (e.g., Always, Almost always, Sometimes, etc.) for each question would be the correct and transparent way to present this data.
Line 212, the discussion correctly identifies “low willingness among healthcare professionals to adopt new communication systems”. However, it then compares the findings to other studies (e.g., a positive view of EMRs in Ghana) without deeply exploring the reasons for the low willingness found in ‘this’ specific study. The data shows significant caution regarding features like multimedia sharing and automated notifications, which is a key finding that deserves more in-depth discussion. Why are the HCPs in this context hesitant? Suggests the discussion should focus more on synthesizing the study’s own results. For example, connect the low internet availability (41.2%) directly to the low enthusiasm for an internet-based system. The narrative should be less about what ‘could be’ with AI and more about what the ‘current reality’ is, as revealed by the data.
Author Response
Healthcare Professionals’ Interactions with Families of Hospitalized Patients Through Information Technologies: Toward the Integration of Artificial Intelligence. Manuscript ID: nursrep-3945818
|
Response to Reviewer 3 Comments
|
||
|
Summary |
|
|
|
Thank you very much for taking the time to review this manuscript. Please find the detailed responses below and the corresponding revisions/corrections highlighted.
|
||
|
Point-by-point response to Comments and Suggestions for Authors |
||
|
Comments 1: The title, abstract, introduction, and conclusion heavily emphasize “Artificial Intelligence”. However, the study itself does not implement, test, or directly measure any aspect of AI. The survey questions pertain to a potential ‘internet-based information system’ with features like multimedia sharing and automated notifications. While these could be ‘components’ of an AI-driven system, they are not inherently AI. The manuscript consistently makes a logical leap from the findings about a basic IT system to broad conclusions about AI integration. Lines 348-351, the conclusion states, “...this vision promotes the development of care ecosystems supported by artificial intelligence... capable of strengthening human-digital interaction...” This is a significant overstatement. The data, which shows low internet access (41.2%) and lukewarm interest in a new system (only ~39% see it as beneficial), points more to fundamental barriers to basic IT adoption than to the readiness for sophisticated AI ecosystems. Suggests the authors should significantly tone down the AI-related claims throughout the manuscript to more accurately reflect the study's scope. The paper is about ‘perceptions of an advanced IT system as a preliminary step toward potential AI integration’. The framing should be adjusted to manage reader expectations and ensure the conclusions are directly supported by the data presented. Response 1: Thank you for pointing this out. We agree with this comment. Therefore, we have addressed your comment on the conclusion in the lines 348-351 by rewriting it more cautiously, in the conclusion section, page ten, last paragraph and line 353. Text added to the manuscript: "On a more general level, this perspective points to the possibility of exploring care environments that gradually and systematically incorporate communication technologies and artificial intelligence-assisted systems, to support specific aspects of human-digital interaction and contribute, in a complementary way, to the care processes that involve patients, their families and health professionals.” Furthermore, claims related to AI have been toned down throughout the manuscript to more accurately reflect the scope of the study, in the abstract section, page one, first paragraph and line 32. Text added to the manuscript: "Preliminary results suggest that the integration of AI-based digital systems in hospital settings remains conditional.” |
||
|
Comments 2: Lines 79-82, the sentence, “Importantly, it underscores the need for flexible tools that respect clinical judgment, empathy, and personalization”, sounds like a finding or a conclusion. The introduction should set the stage and state the research question, not pre-emptively declare the study’s findings. Suggests this sentence should be rephrased or moved to the Discussion/Conclusion section, where it would be appropriate to interpret the study’s results. Response 2: Thank you for pointing this out. We agree with this comment. Therefore, we have addressed your comment, rewriting the sentence according to the introduction section, this is also indicated, page two, paragraph three in the introduction section and line 79. Text added to the manuscript “It addresses the need for flexible tools that adapt to clinical criteria and preserve fundamental elements of care, such as empathy and personalization.” Comments 3: Lines 97-100, the justification for the sample size is weak. Stating it is “based on the number of participants in pilot studies [19]” is vague. While the mention of a 5:1 to 15:1 ratio of sample-to-variable is a known rule of thumb for factor analysis, the authors do not state how many variables were used in their analysis to demonstrate that this ratio was met. Suggests: while this is a pilot study and a small sample is expected, the justification could be clearer. More importantly, the limitations section already correctly notes the small N, which mitigates this issue to some extent. However, the initial justification in the methods section should be phrased more cautiously. Response 3: Thank you for pointing this out. We agree with this comment. Therefore, we have addressed your comment, the justification for the sample size is weak, this is also indicated, page three, first paragraph in the materials and methods section and line 100. Text added to the manuscript “The sampling was subject to the availability of the respondents' workload. This made it impossible to apply a probabilistic procedure. Therefore, convenience sampling was chosen, a strategy frequently used in exploratory, preliminary and pilot studies.” Additionally, a paragraph from the discussion section was moved to preface text to point out that the study was conducted during the pandemic, making it difficult to access the population to obtain the probabilistic sample. This is also indicated, page three, first paragraph in the materials and methods section and line 96. Text added to the manuscript “It should be noted that the study was conducted during the Severe Acute Respiratory Syn-drome Coronavirus 2 (SARS-CoV-2) pandemic, which posed difficulties in recruiting participants for the study [19], and the main barriers were health restrictions and protocols, staff burnout, and turnover [20].” It also specifies how many variables were used in the analysis to demonstrate that this proportion was met. This is indicated in the same section, page and paragraph, on line 103. Text added to the manuscript “The instrument used in this study comprises three latent dimensions—Use of Information Technologies, Information Needs, and Interactions—which were treated as the primary analytical variables [22].” Comments 4: Line 140-198, results section, the results are reported with two decimal places (e.g., 58.80%, 37.30%). For a small sample size of N=51, this implies a level of precision that does not exist. For instance, 19 out of 51 participants is 37.2549...%. Reporting this as 37.30% is statistically inappropriate. Suggests percentages should be rounded to one decimal place (e.g., 37.3%) or to the nearest whole number (e.g., 37%), which is standard and more honest for this sample size. Response 4: Thank you for pointing this out. Therefore, we have addressed your comment, the percentages have been rounded to one decimal place throughout the work. *Comments 5: Line 162-163, 173-174, 190-191, Figures 1, 2, and 3 are highly problematic and present the data in a misleading way. They selectively display only one response category (e.g., “Always” or “Only sometimes”) for each question, omitting the rest of the response distribution (e.g., “Almost always”, “Almost never”, “Never”). This practice is inappropriate as it hides the full context of the participants’ responses and can lead to incorrect interpretations. For example, in Figure 1, stating that 41.20% “always” use the internet does not tell the reader what the other 58.80% do. Suggests the bar charts must be revised to show the full distribution of responses for each item on the Likert scale. A stacked bar chart or grouped bar chart showing the percentages for all categories (e.g., Always, Almost always, Sometimes, etc.) for each question would be the correct and transparent way to present this data. Response 5: Thank you for pointing this out. We agree with this comment. Therefore, we have addressed your comment, in response to your comment, Figures 1, 2, and 3 have been redesigned as stacked bar charts showing the full distribution of responses to each question. This is also indicated, pages five to seven, in the results section and lines 166, 176 and 194. Comments 6: Line 212, the discussion correctly identifies “low willingness among healthcare professionals to adopt new communication systems”. However, it then compares the findings to other studies (e.g., a positive view of EMRs in Ghana) without deeply exploring the reasons for the low willingness found in ‘this’ specific study. The data shows significant caution regarding features like multimedia sharing and automated notifications, which is a key finding that deserves more in-depth discussion. Why are the HCPs in this context hesitant? Suggests the discussion should focus more on synthesizing the study’s own results. For example, connect the low internet availability (41.2%) directly to the low enthusiasm for an internet-based system. The narrative should be less about what ‘could be’ with AI and more about what the ‘current reality’ is, as revealed by the data. Response 6: Thank you for pointing this out. We agree with this comment. Therefore, we have addressed your comment, the limited use of information technologies by healthcare professionals and their reluctance to adopt new communication systems are explored. This is also indicated, page seven, paragraph two in the discussion section and line 217. Text added to the manuscript “This scenario suggests that a lack of familiarity with IT and the perception that its incorporation may complicate workflows or increase workload negatively influence adoption, while organizational factors such as insufficient training, limited technical support, and a lack of institutional incentives also appear to reinforce this resistance [31].” Additionally, the opportunity is addressed that the quality of the information provided exceeded average expectations. This is also indicated, page seven, paragraph two in the discussion section and line 222. Text added to the manuscript “This fact demonstrates an untapped potential: current systems offer clinical value, but their impact depends on implementation strategies that improve usability, strengthen digital skills, and highlight practical benefits for healthcare professionals [32].” |
||
